# High-resolution neutron imaging of salt precipitation and water transport in zero-gap $CO_2$ electrolysis

Joey Disch [1,2], Luca Bohn [1,2], Susanne Koch [1,3], Michael Schulz [4], Yiyong Han[4], Alessandro Tengattini [5,6], Lukas Helfen [6], Matthias Breitwieser[1,3] & Severin Vierrath [1,2,3]

The electrochemical reduction of $CO_2$ is a pivotal technology for the defossilization of the chemical industry. Although pilot-scale electrolyzers exist, water management and salt precipitation remain a major hurdle to long-term operation. In this work, we present high-resolution neutron imaging (6 μm) of a zero-gap $CO_2$ electrolyzer to uncover water distribution and salt precipitation under application-relevant operating conditions (200 mA cm$^{-2}$ at a cell voltage of 2.8 V with a Faraday efficiency for CO of 99%). Precipitated salts penetrating the cathode gas diffusion layer can be observed, which are believed to block the $CO_2$ gas transport and are therefore the major cause for the commonly observed decay in Faraday efficiency. Neutron imaging further shows higher salt accumulation under the cathode channel of the flow field compared to the land.

While there are clear pathways for the defossilization of the energy or transport sector, solutions are still needed for other sectors, which rely on fossil feedstock like oil and natural gas[1]. One technology that emerged as a promising candidate to provide fossil-free feedstock to the chemical industry is the electrochemical reduction of (captured) $CO_2$[2]. Many different reactor designs and target products have been proposed for this application, ranging from $C_1$ to $C_3$ products like carbon monoxide, formic acid, ethylene, and 1-propanol[3,4]. Gas-fed zero-gap electrolyzers comprising an anion exchange membrane (AEM) are currently the most promising reactor design to meet the requirements for industrial commercialization[5] and electrolyzers having carbon monoxide (CO) as target product are already reaching pilot scale[6]. In spite of reported peak partial current densities of up to 1.0 A cm$^{-2}$[7] and Faraday efficiencies (FE) for CO above 90% (at lower current densities), these types of electrolyzers still face major challenges regarding operation stability and durability[8,9]. Especially electrode dry out and flooding events, as well as salt precipitation are

often made responsible for performance loss or even cell failure in such systems[10-13]. However, a clear understanding of these limiting effects is still missing.

The main reactions and transport mechanisms during electrochemical $CO_2$ reduction in alkaline environments are displayed in the schematic drawing in Fig. 1a. The $CO_2$ reduction reaction on the cathode side as well as the competing hydrogen evolution reaction (HER) consume water (reaction I–III).

Cathode reactions:

$$CO_2 + H_2O + 2e^- \rightarrow CO + 2OH^- \qquad (I)$$

$$2H_2O + 2e^- \rightarrow H_2 + 2OH^- \qquad (II)$$

$$CO_2 + H_2O + 2e^- \rightarrow HCOO^- + OH^- \qquad (III)$$

[1]Electrochemical Energy Systems, IMTEK - Department of Microsystems Engineering, University of Freiburg, Georges-Koehler-Allee 103, 79110 Freiburg, Germany. [2]University of Freiburg, Institute and FIT – Freiburg Center for Interactive Materials and Bioinspired Technologies, Georges-Köhler-Allee 105, 79110 Freiburg, Germany. [3]Hahn-Schickard, Georges-Koehler-Allee 103, 79110 Freiburg, Germany. [4]Heinz Maier-Leibnitz Zentrum (MLZ), Technische Universität München, Garching, Germany. [5]Univ. Grenoble Alpes, CNRS, Grenoble INP, 3SR, 38000 Grenoble, France. [6]Institut Laue-Langevin, 71 avenue des Martyrs - CS 20156, 38042 Grenoble, France. e-mail: Severin.Vierrath@imtek.uni-freiburg.de

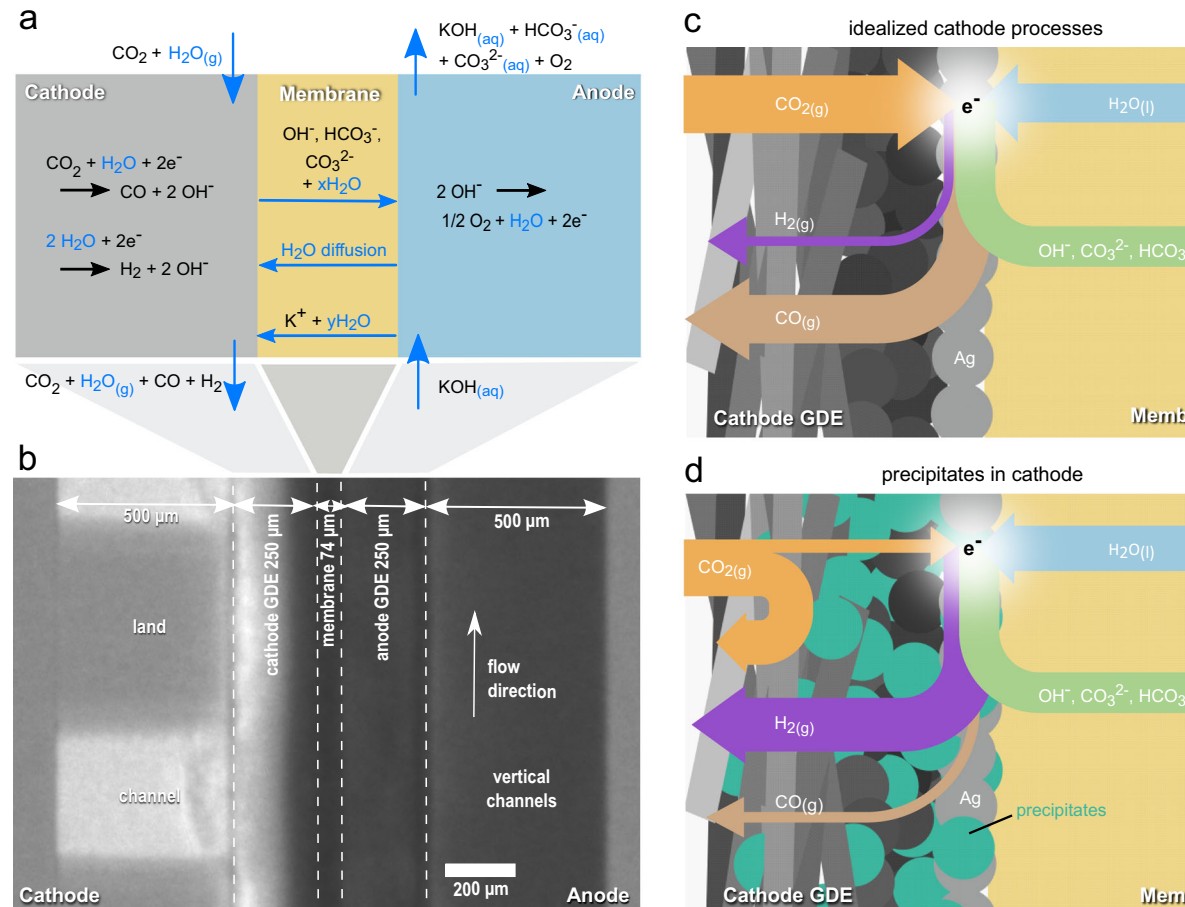

**Fig. 1 | Transport processes and cell components. a** water management high-lighted in schematic of reactions and transport processes during operation. **b** enlarged section of a normalized neutron radiographic image of the $CO_2$ electrolysis cell with approximate dimensions of the components. **c** idealized cathode processes. **d** influence of precipitates on the cathode processes.

By humidifying the $CO_2$ gas stream, water is introduced to the cathode compartment. Through a concentration gradient of water between cathode and anode, additional water is transported from the anode side to the cathode side by diffusion. The hydroxide ions produced in the cathode reactions rapidly react with the gaseous $CO_2$ forming bicarbonate ($HCO_3^-$) and carbonate ($CO_3^{2-}$) ions in the following carbonation reactions:

$$CO_2 + OH^- \rightarrow HCO_3^- \tag{IV}$$

$$HCO_3^- + OH^- \rightarrow CO_3^{2-} + H_2O \tag{V}$$

Jeng et al. showed that bicarbonate and carbonate ions are thereby the primary charge carriers passing the membrane, dragging water in their hydration shell with them from the cathode to the anode compartment[14]. On the anode side, hydroxide ions are oxidized producing oxygen and water.

Anode reaction:

$$2\,OH^- \rightarrow 1/2\,O_2 + H_2O + 2\,e^- \tag{VI}$$

Although the AEM hinders potassium cations from crossing the membrane in bigger quantities by Donnan exclusion, a small amount of cations is still migrating through the membrane. As cations move from the anode to the cathode side, they drag water in their hydration shell with them. Over time potassium, carbonate and bicarbonate ions can accumulate at the cathode. When exceeding the solubility limit,

by increasing ion concentration or by consumption of water, salt crystallites can precipitate on the cathode side. Figures 1c, d illustrates how precipitates supposedly influence the processes in the cathode gas diffusion electrode (GDE). Carbonate precipitates are normally porous and hydrophilic, increasing the capillary pressure inside the cathode GDE[15]. Hence, the precipitates make the GDE prone to electrode flooding. Liquids and precipitates can clog pores in the catalyst layer, microporous layer and the gas diffusion layer of the cathode GDE, and thereby impede the gas transport[11]. With an increasing amount of precipitates, the mass transport resistance for the $CO_2$ to reach the active sites of the catalyst layer increases and the $CO_2$ concentration at the catalyst surface will eventually decrease. The lower $CO_2$ concentration may cause a selectivity shift towards the HER, as water can still be sufficiently supplied from the anode side through the membrane.

In situ neutron radiography has proven to be a powerful tool to investigate water transport in multiple electrochemical applications, such as fuel cells or water electrolyzers[16,17]. The technological advances of neutron imaging detectors significantly improved the resolution of neutron radiography in recent years. Nonetheless, only two neutron-imaging studies investigating $CO_2$ electrolyzers have been published up to now. The first study investigated bubble formation in a liquid buffer layer on the cathode side[18]. Besides high cell potentials, cells with a liquid catholyte between the membrane and cathode GDE often exhibit fast electrode flooding, as the cathode is in direct contact with the liquid electrolyte. The circulating electrolyte, however, can wash away precipitates and depending on its chemical composition change the ion concentration at the cathode side compared to a zero-gap cell.

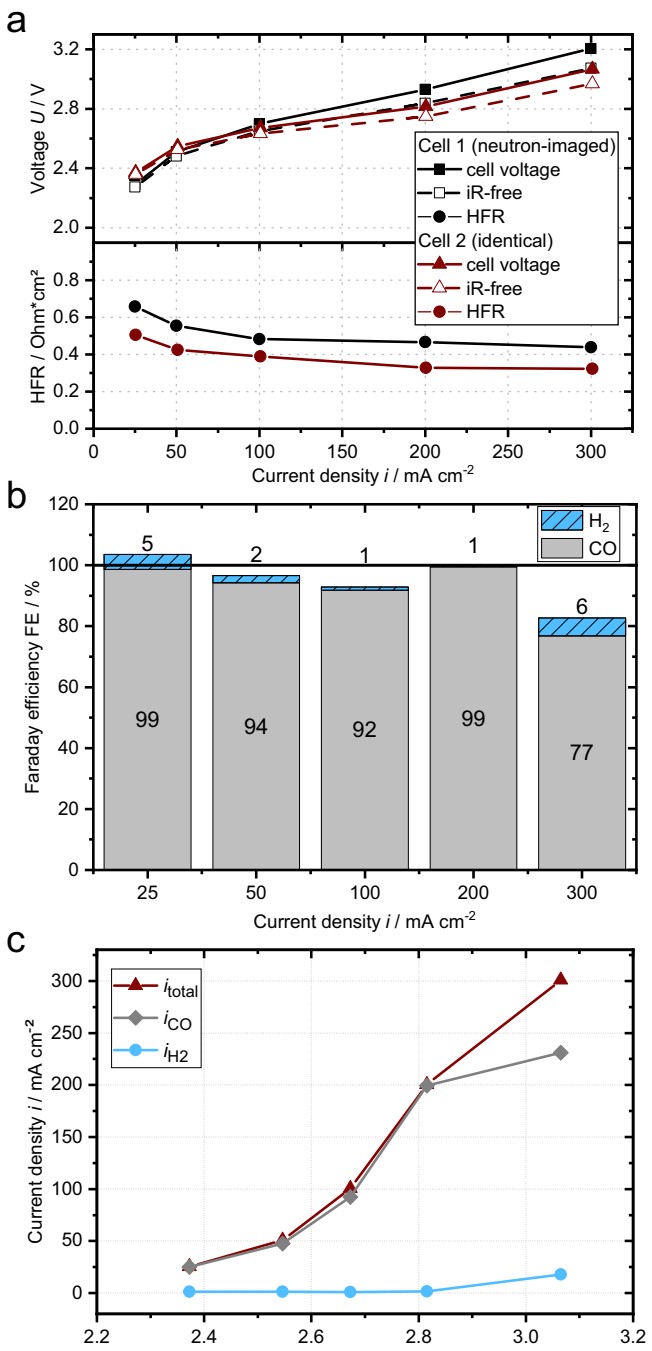

**Fig. 2 | Electrochemical performance of the investigated CO$_2$ electrolysis cell.**
**a** cell voltage, iR-free cell voltage and high-frequency resistance (HFR) over current density. **b** Faraday efficiencies of CO and H$_2$ of cell 2. **c** partial current densities of CO and H$_2$ (cell 2). The cell comprised a Sustainion® membrane, an IrO$_2$ anode and an Ag cathode (all purchased from Dioxide Materials™). Measured at ambient temperature (23 ± 3 °C), with a bubbler at 50 °C humidifying the cathode gas stream (47.5 sccm CO$_2$ and 2.5 sccm N$_2$) and 0.1 KOH anolyte circulated at 20 mL min⁻¹. Source data are provided as a Source Data file.

The second study conducted neutron imaging of a zero-gap cell but with a cation exchange membrane[19]. Zero-gap cells using cation exchange membranes show significantly lower CO selectivity compared to cells using AEMs, as they provide an acidic reaction environments to the cathode, facilitating the competing HER[20]. Furthermore, protons moving from the anode to the cathode are responsible for the ionic charge transfer between the anode and cathode. The ionic

transport and the electroosmotic drag are directed in the opposite direction compared to cells using AEMs, drastically changing the water balance. As the water balance and flooding behavior in the above-mentioned studies strongly differ, their results cannot be applied to AEM-based zero-gap CO$_2$ electrolyzers. Etzold et al. have recently highlighted this lack of neutron studies for zero-gap CO$_2$ electrolysis in a review[21]. This work therefore uses high-resolution operando neutron radiography to elucidate the water transport in an alkaline zero-gap CO$_2$ electrolysis cell converting gaseous CO$_2$ to CO.

## Results and discussion
### Electrochemical performance
Figure 2 shows the electrochemical performance of two identical cell assemblies with a Sustainion® membrane, an IrO$_2$ GDE anode and an Ag GDE cathode (all purchased from Dioxide Materials™). As can be seen from Fig. 2a, the first cell, which was neutron-imaged, had a slightly higher high-frequency resistance (HFR) than the second cell (average ΔHFR ≈ 0.1 Ω cm²) and thus a generally higher voltage (e.g. 2.9 V vs. 2.8 V at 200 mA cm⁻²). This could be caused by different durations of membrane storage in potassium hydroxide solution, variance in temperatures during the experiments (lab vs. beamline) or slight inaccuracies during cell assembly.

The corresponding CO Faraday efficiencies (FE$_{CO}$) measured for cell 2 stay above 92% between 25–200 mA cm⁻² with a maximum FE$_{CO}$ of 99% at 200 mA cm⁻² (Fig. 2b). At higher current densities (300 mA cm⁻²) the FE$_{CO}$ drops to 77% while the hydrogen evolution accounts for 6% of the product gas. Yet, the rise in FE$_{H2}$ does not fully compensate for the drop in FE$_{CO}$ resulting in a combined FE$_{total}$ = FE$_{CO}$ + FE$_{H2}$ of 83%. As seen in Fig. 2c, for 25–200 mA cm⁻² the partial current density for CO does only slightly differ from the total current density. However, at 300 mA cm⁻² approximately 51 mA cm⁻² is going into other faradaic processes, with products that are not detected by the gas chromatography, potentially formate (HCOO⁻)[10,22]. Despite small deviations in cell voltage and HFR, both cells show a similar electrochemical performance in a range relevant for industrial commercialization ($i$ > 200 mA cm⁻² below 3 V and FE > 90%[23–25]).

### Water transport and salt precipitation
Representative normalized neutron radiographic images of the electrolyzer at different operation points are shown in Fig. 3a. On the cathode side (left) channels and gas diffusion layer (GDL) appear bright (lower attenuation) as they are filled with gaseous CO$_2$, while the anode side (right) is generally darker due to the aqueous KOH solution in the channel and GDE (see Fig. 1b for a detailed labeling of all cell parts). At 0 mA cm⁻² the corners and edges of the cathode flow field are wetted by condensing water, visible as a dark shading along the edges. With increasing current density, the water consumption of the cathode reactions rises and the shading in the cathode channel gets gradually lighter and at 200 mA cm⁻², the condensate has fully dried off.

Dark dents on the backside of the cathode GDE are reaching into the flow field channels and are penetrating the gas diffusion layer (a representative dent is marked with * in Fig. 3a). Their attenuation reduces with rising current density, indicating drying in that area. Nevertheless, at 300 mA cm⁻² these bulges are still clearly visible, whereas the liquid condensate in the channel has fully dried off. Even though the images are averaged over several minutes, the bulges show sharp contours, indicating an immobile solid phase. The precipitation of carbonic salts is a widely known problem in zero-gap AEM CO$_2$ electrolyzers. Various groups reported that after the experiments, precipitates in flow channels and on the backside of the GDE were even visible to the bare eye[11,26–29]. Kong et al. recently presented a study, quantifying the distribution of potassium carbonate (K$_2$CO$_3$) and potassium bicarbonate (KHCO$_3$) in the GDEs post-mortem and showing precipitated salts penetrating the entire depth of the GDEs[30]. Endrődi et al. made similar observations[31]. K$_2$CO$_3$ shows significantly

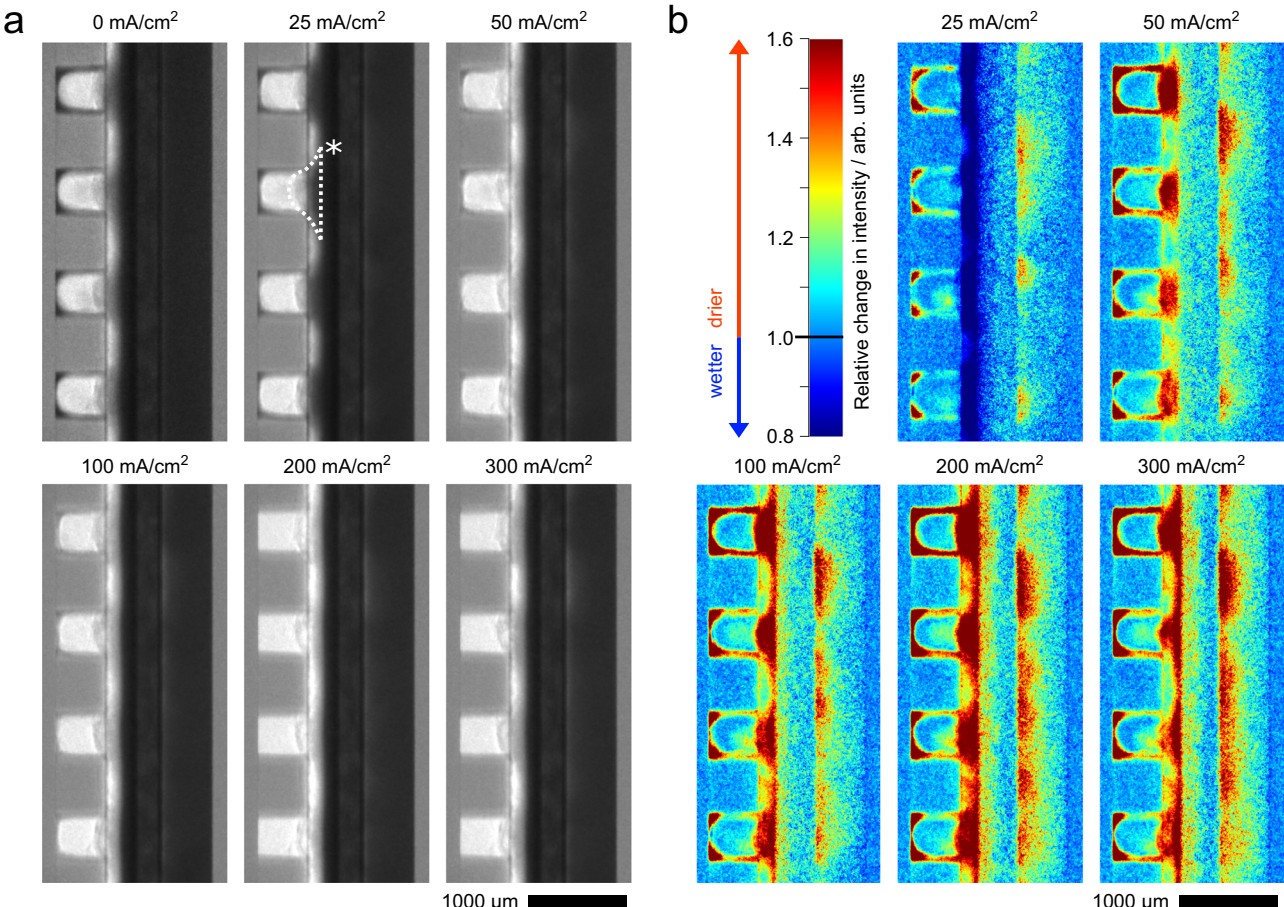

**Fig. 3 | Neutron radiography of the CO₂ electrolysis cell. a** Averaged neutron radiographic images at different current densities. * marks a dark area potentially caused by precipitates. **b** The relative change in intensity normalized to the intensity of the zero current cell, which corresponds to a change in water content.

lower neutron absorption and scattering rates than water (see Supplementary Table 1 and Supplementary Fig. 6 for neutron transmission rates). Due to its low neutron attenuation, $K_2CO_3$ can only be observed indirectly, by imaging the retained water. Considering the deliquescent properties of $K_2CO_3$ and the moist reaction environment (humidified cathode gas stream, aqueous anolyte, a membrane with a high water uptake[32] and low temperatures $23 \pm 3\,°C$) carbonates inside the cell are very likely hydrated[15]. This should be the case, especially in the zero current state, when no water is consumed by the cathode reactions and condensing water in the cathode flow field is observed.

KHCO₃ instead contains a hydrogen atom in its structure, yielding higher interactions with neutrons than $K_2CO_3$. To some extent, even dry KHCO₃ salt should be visible in neutron radiography (see Supplementary Fig. 6). Nonetheless, KHCO₃ is also strongly hygroscopic and retains water, which might more drastically affect the neutron opacity in a humid environment. Post-operation scanning electron microscopy and energy dispersive X-ray spectroscopy (SEM/EDX) measurements of a cathode cross section further show precipitates containing potassium in the silver catalyst layer, the micro-porous layer and in the gas diffusion layer (see Supplementary Fig. 7). The presence of KHCO₃ in the catalyst layer was additionally verified by post-operation Raman measurements (see Supplementary Fig. 8). Considering those results and the mentioned observations made by other groups[30,31], it can be assumed that the dark dents originate from precipitated KHCO₃ and water retained by the precipitates in that area.

By that, the intensity differences alongside the cathode GDE allow us to deduce where precipitates are mainly located in the electrode. For example, as the cathode GDE is brighter in the land area of the cathode flow field even at 0 mA cm⁻², it can be assumed, that less

precipitates are located in that area (note that KHCO₃ and $K_2CO_3$ might have accumulated in the cathode GDE already in the preceding electrochemical measurements described in the methods section). The brightening of the dark areas with rising current density indicates drying either by consumption of water or by its transportation to the anode via electro-osmotic drag (see Fig. 1a).

On the anode side, a slight brightening, especially at the interface between GDE and channel can be observed with increasing current density. Due to the poor contrast in that area, further evaluation of those images is difficult. Therefore, all images were normalized with respect to the cell at zero current by dividing the image of interest by the zero current image (Fig. 3b). Values below one indicate a decrease in intensity and therefore an increase of water content compared to the zero current state. Values greater than one indicate drying of that region. These normalized images show the afore-mentioned observations more clearly. With rising current density, the oxygen evolution on the anode side leads to visibly higher accumulation of gas bubbles in the anode flow channels. In addition, spatial differences of the bubble accumulation alongside the flow channel can be observed. On the cathode side, the relation of water distribution and applied current density is less univocal. While at 25 mA cm⁻² the condensate accumulation at the edges of the flow channels is reduced, the cathode GDE is slightly wetter than without current. At 50 mA cm⁻², the whole cathode starts to become dryer, while the degree of dry-out significantly differs between channel and land area of the cathode GDE. With increasing current densities, the whole cathode GDE is gradually drying. Especially the region with precipitates and a thin layer close to the membrane become dryer. Going from 200 to 300 mA cm⁻² this trend for the cathode GDE is coming to halt and the intensity in some regions of the

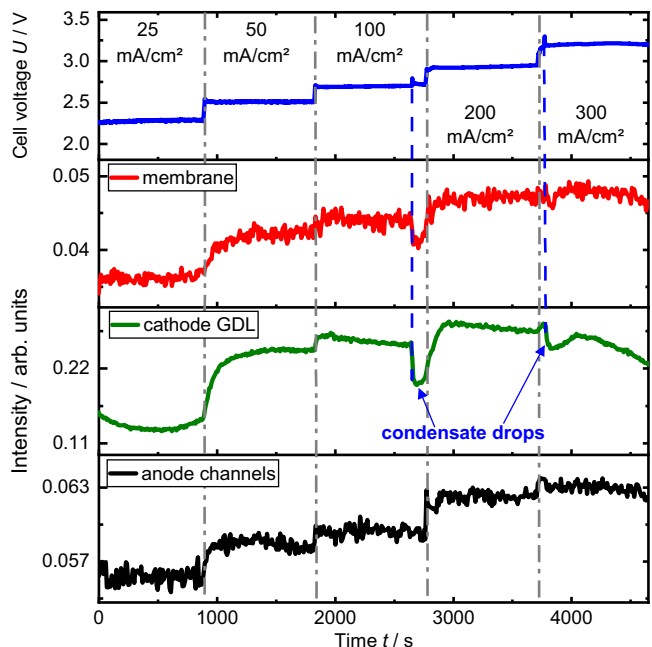

**Fig. 4 | Temporal evolution of the neutron imaging experiment.** Cell voltage and averaged intensities of membrane, cathode gas diffusion layer (GDL) and anode channels area over time. The change in intensity values corresponds to a change in water or precipitate content. See Supplementary Fig. 4 for selected regions. Source data are provided as a Source Data file.

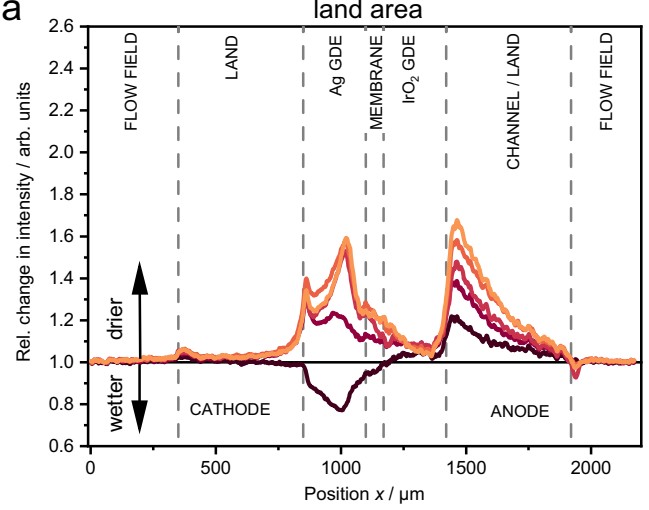

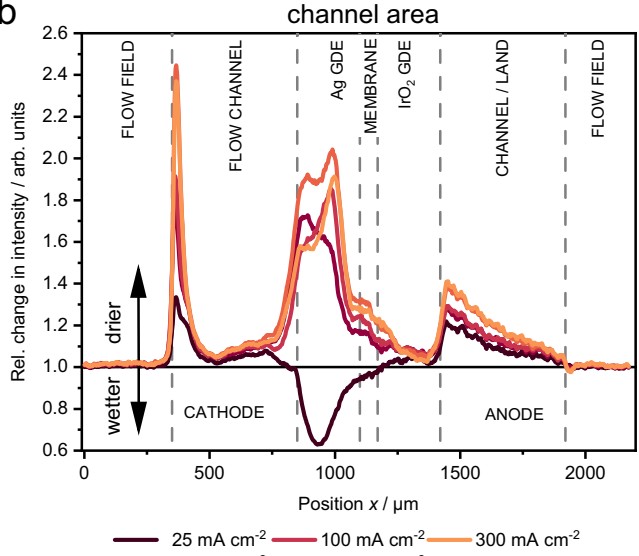

**Fig. 5 | Comparison of channel and land area.** Relative change in intensity at different current densities compared to open circuit in **a** cathode flow field land area, and **b** cathode flow field channel area. A value larger than one indicates less water/$KHCO_3$ and a value below one more water/$KHCO_3$, as compared to the zero current reference. See Supplementary Fig. 5 for selected regions. Source data are provided as a Source Data file.

cathode gas diffusion layer even decreases. As the water consumption significantly increases from 200 to 300 mA cm$^{-2}$, it is unlikely that water causes this drop in intensity. However, besides water, two other processes could be responsible for that intensity drop: the formation of either $KHCO_3$ or potassium formate (HCOOK) salts.

Weng et al. simulated the processes in a zero-gap $CO_2$ electrolysis cell. The simulations predict that the solubility limit of carbonates produced at the cathode is reached at approximately 750 mA cm$^{-2}$ [33]. Under less ideal conditions, this solubility limit can already be reached at significantly lower current densities, as shown by Wheeler et al. (visible precipitation at 100 mA cm$^{-2}$)[29]. The electrochemical reduction of $CO_2$ to HCOO$^-$, as depicted in Eq. (III), has been reported by several groups as an occurring side reaction on silver catalysts in $CO_2$ electrolysis cells[10,22]. Considering the drop in Faraday efficiency at 300 mA cm$^{-2}$ and the potentially high local alkalinity at the electrode surface, it can be assumed that HCOO$^-$ is produced also in this case[34].

Both HCOOK and $KHCO_3$ salts show similar neutron attenuation and high concentrations should be visible in neutron radiography (see Supplementary Fig. 6 for neutron transmission rates). By only looking at the radiographic images, it is not possible to differentiate between the contribution of water or the different salts to the intensity decrease. However, considering the afore-mentioned points and the post-operation Raman and SEM/EDX measurements (see Supplementary Figs. 7 and 8), it is most probable that the accumulation of $KHCO_3$ causes the observed drop in intensity.

To better discriminate potential non-stationary effects, Fig. 4 displays the time evolution of the cell voltage and of the neutron image intensity of the membrane including electrode layers, the cathode gas diffusion layer and the anode flow channels (see Supplementary Fig. 4 for selected regions). The cell voltage is stable during each current step ($\Delta U < 2.0$ mV min$^{-1}$), with the exception of two small spikes at 100 mA cm$^{-2}$ and 300 mA cm$^{-2}$, most probably caused by condensate from the gas humidification entering the cathode inlet. By increasing the water content in the cathode, the condensate droplets lead to a decrease in intensity, visible as intermediate intensity drop in the

cathode GDL and membrane area. From 25 to 200 mA cm$^{-2}$ all intensity values are on average increasing with each current step, indicating higher bubble accumulation on the anode side and more water consumption on the cathode side. This coincides with the observations made in Fig. 3. However, after an initial rise of the cathode GDL intensity at 100 and 200 mA cm$^{-2}$, the intensity is continuously decreasing with a similar rate for both currents. Reaching 300 mA cm$^{-2}$, the intensity starts decreasing more drastically, even falling below the average intensity measured at 50 mA cm$^{-2}$. Moreover, at the end of the 15 min period, the intensity has not yet reached a plateau.

Again, if the change in intensity would only account for water content, this observation would be counterintuitive, as the increased reaction rate leads to higher water consumption and electro-osmotic drag from cathode to anode (see Fig. 1a). This observation in the cathode GDL suggest that $KHCO_3$ is continuously accumulating during cell operation. The different slopes of the cathode GDL intensity curve translate to $KHCO_3$ accumulation already starting at 100 mA cm$^{-2}$ and significantly increasing at 300 mA cm$^{-2}$.

As clear differences between channel and land area were observed in cathode drying and anode gas formation (Fig. 3), the spatial differences are studied in more detail. Figure 5 shows the relative change in intensity along the cell profile, for each current density relative to the zero current cell. Each curve represents the average of three land and three channel profiles (see Supplementary Fig. 5). The drying of the cathode GDE is more pronounced in the channel area, which indicates that the $CO_2$ gas stream is generally removing more water than bringing into the cell. This implies a net water flux from anode to cathode, which is probable in the current conditions of a fully humidified gas in the cathode and liquid 0.1 M KOH in the anode[29].

Figure 5 also shows that the bubble accumulation on the anode side is more pronounced in the region of the cathode land, indicating higher local currents in the cathode land region. This implies that ions or electrons and not gaseous reactants generally limit the cathode reaction, since the land region has a higher current despite the typically lower gas concentration. This is in line with the electrochemical data, which does not feature the typical mass transport limitation that causes an overpotential, bending the IV-curve above the linear relation (Fig. 2a)[35]. While this might be surprising, one has to consider that a region depleted of $CO_2$ might still have a sufficiently high water content for the hydrogen evolution reaction. The utilized Sustainion® membranes were reported to have a very high water uptake of ~80%[32,36] and are only getting slightly drier during cell operation (see Fig. 5), directly providing water to cathode catalyst layer. $CO_2$, however, has to reach the active sites of the cathode catalyst layer by diffusing from the cathode flow field channel through the gas diffusion layer, the microporous layer and finally the pores of the catalyst layer. With rising current density, more $CO_2$ is consumed by the $CO_2$ reduction reaction and the carbonation reaction, lowering the $CO_2$ concentration at the electrode surface. Precipitates in the electrode add an additional diffusion barrier for $CO_2$ (see Fig. 1d). Thus the reaction shifts to the HER resulting in lower faradaic efficiency of the $CO_2$ reduction at high current densities (Fig. 2c) instead of increasing the overpotential.

The previously mentioned wetting of the cathode GDE at 25 mA cm$^{-2}$ is also easily distinguished in this graph for both channel and land. In line with the current understanding of electro-osmotic drag, the hydroxide ions drag along water, humidifying the cathode and even slightly the membrane. At higher currents, however, the cathode and membrane are dryer than in the zero current state, which indicates that water consumption and electro-osmotic drag are higher than water-sorption/resupply from the gas phase and diffusion from the anode.

Although precipitates and associated electrode flooding are visible in the neutron images over the entire current range (precipitates potentially already formed during pretests), the cell is demonstrating good cell performance with FE$_{CO}$ above 90% for current densities below 300 mA cm$^{-2}$. This indicates that gas transport is still sufficient in that region. Only reaching 300 mA cm$^{-2}$, when HCOO$^-$ formation and salt precipitation are increasing significantly, the FE$_{CO}$ decreases. This shows that the electrochemical data alone is a poor indicator for the presence of precipitates.

Several strategies have been proposed in literature to mitigate salt precipitation on the cathode side[11,26,27,31], but no approach has emerged as the most promising so far. Recent publications reporting high partial current densities for CO are most often operating at elevated temperatures of up to 65 °C[7,37]. Therefore, after the replication of the neutron imaging experiments, further measurements at a cell temperature of 50 °C were conducted with the identical cell 2 (see Supplementary Fig. 3). The elevated temperature facilitates enhanced $CO_2$ reduction performance (FE$_{CO}$ of 99% at 300 mA cm$^{-2}$ and 2.8 V). The cell reaches a maximum in CO partial current density of 435 mA cm$^{-2}$ and the FE$_{CO}$ starts decreasing at 500 mA cm$^{-2}$. The positive effects of the elevated temperature could be due to a multitude of factors like improved kinetics, enhanced mass transport, higher solubility or ion conductivity.

In summary, this work investigates the spatio-temporal evolution of water distribution in a gas-fed, zero-gap $CO_2$ electrolyzer cell at different operation points by employing high-resolution neutron radiography. The cell shows industrially relevant cell performance (200 mA cm$^{-2}$ at a cell voltage of 2.8 V and FE$_{CO}$ of 99%). Even though intrinsic material restrictions (wet assembly of Sustainion®) or generation of side products with high neutron interaction complicate the quantification of water in $CO_2$ electrolyzers compared to water-based applications like fuel cells or water electrolyzers, several important findings can be derived: A drop in faradaic efficiency at 300 mA cm$^{-2}$ was observed, indicating HCOO$^-$ formation. Precipitated salts and precipitate-associated electrode flooding in the cathode GDE are visible throughout the entire experiment. At higher current densities, the accumulation of precipitates further increases. The water distribution on the cathode side is heavily influenced by the hygroscopic properties of the precipitated salts. The high spatial resolution makes these small structural changes apparent on a scale of some μm. More precipitates are visible in the cathode channel area than in the land area. Furthermore, bubble accumulation was clearly visible on the anode side. The observed local differences suggest higher local reaction rates in the cathode land region, which indicates that the overall cathode reaction is not limited by gaseous transport up to 300 mA cm$^{-2}$ but is rather shifting to hydrogen evolution reaction when $CO_2$ is depleting. Furthermore, higher local reaction rates can lead to local dry-out and hotspots potentially leading to ionomer degradation and finally cell failure. The influence of flow field design and local cell compression should therefore be included in future development to further increase the performance and durability of $CO_2$ electrolyzers. The same applies for the mitigation of the observed salt precipitation in AEM $CO_2$ electrolyzers. In summary, this work demonstrates that high-resolution neutron radiography is a powerful tool to investigate in-situ processes like water transport as well as salt precipitation and local current density distributions in gas-fed $CO_2$ electrolyzers.

## Methods
### Materials
A zero-gap electrolysis cell (2 cm × 1 cm active cell area) was custom-built (see Supplementary Fig. 1 and 2 for schematics of the experimental setup). The cell fixture is designed for in-plane neutron imaging. It consists of two stainless steel (AISI 904 L) end plates and two flow fields made from titanium grade 2.0 coated in gold. A serpentine flow field design has been chosen for the gas flow on the cathode side, whereas the anode comprises a vertical parallel flow field. Figure 1b shows an enlarged section of a representative neutron radiography, labeling the different cell components and their corresponding thicknesses. The flow channels on either side are 500 μm wide and deep. The utilized silver cathode GDE, the $IrO_2$ anode GDE (both described under Dioxide Materials™ US patent 9,555,367) as well as the Sustainion® X37-50 RT anion exchange membrane were purchased from Dioxide Materials™. The received electrodes (5 cm × 5 cm) were cut into smaller pieces (2 cm × 1 cm). They were then compressed to a thickness of 500 μm (~30% compression) with two 250-μm PTFE gaskets acting as "hard stop". Custom punching tools were used to reproducibly cut gaskets and electrodes. Two centering pins ensure precise alignment of all fixture components during assembly. The membrane thickness was measured ex situ with a micrometer gauge (74 μm) and is assumed incompressible. Before cell assembly, the membrane was ion exchanged by immersing it in a 1 M potassium hydroxide solution (made from ACS reagent ≥85% pellets, Sigma-Aldrich and deionized water) for 24 h and was then stored in a sealed container filled with fresh 1 M KOH solution until needed. The ion exchange of the electrodes, also in 1 M KOH, was conducted approximately 12 hours before the cell measurements. Before stacking the electrodes and membrane, excess liquids were carefully removed with paper tissue. The cell was mounted onto an electronically controlled

motion stage for precise alignment of the cell between the detector and neutron beam. The neutron beam enters through an in-cut window of the flow fields. In the window region the flow fields measure 2 cm in beam direction and the active cell area measures 2 cm vertically and 1 cm horizontally (in beam direction).

## Electrochemical measurements

The cathode was supplied with a mixture of 47.5 sccm $CO_2$ (grade 4.5, Messer Industriegase GmbH) and 2.5 sccm Nitrogen (grade 5.0, Linde) by two digital mass flow controllers (Bronkhorst). A bubbler heated to 50 °C was used to humidify the cathode gas stream. 0.5 L of a 0.1 M KOH solution was circulated on the anode side at a flow rate of 20 mL min$^{-1}$ with a peristaltic pump (Reglo ICC 7800-58, Ismatec). A Biologic Galvono-/Potentiostat VSP-300 was used for all electrochemical measurements. The cell performance was evaluated by applying different current densities from 0 mA cm$^{-2}$ to 300 mA cm$^{-2}$. Each current was applied for 15 minutes and at the end of each current step, a galvanostatic electrochemical impedance spectroscopy measurement (1–500 kHz and 5% amplitude) was conducted to determine the high-frequency resistance. Gas compositions were quantified with an Agilent micro gas chromatograph (GC) 990 equipped with a CP-COX column. The Faraday efficiencies were only determined using an identical cell (electrodes and membranes from the same batch) with the same setup and measurement procedure as for the imaging experiments. GC measurements were conducted after 14 minutes of each current step.

After cell assembly the cell was put on the holder and connected to the tubing as quickly as possible to avoid drying of the cell. Gas flow and anolyte flow were started immediately after connecting the tubing. The cell was flushed with electrolyte for approximately 10 min before starting the first measurements. Before the presented measurements, several pretests were conducted. The pretest protocol starts with several voltages steps from 1.5 to 3.2 V of 50 s each followed by an EIS measurement and a linear sweep voltammetry. Immediately after that, the first current step experiment was started with a bubbler temperature of 30 °C. Due to the very short pause between the pretests and the current steps, the cell did not have enough time to reach an equilibrated state. The effects on the water content of the pretests were superposing the actual effects of the low current densities (25–50 mA cm$^{-2}$) applied in the current step experiment. Therefore, that data was not evaluated.

## Electrochemical calculations

The cell potential corrected by the contribution of ohmic overpotential ($U_{iRfree}$) was calculated as follows:

$$U_{iRfree} = U_{cell} - i \times HFR_i.$$

$U_{cell}$ stands for the measured cell potential, $i$ for current density and $HFR_i$ for the high frequency resistance at the specific current density. The HFR was determined by taking the x-axes intercept of a Nyquist plot of the EIS measurements conducted at each current density.

The Faraday efficiency for the product $x$ was calculate using this equation

$$FE_x = n_x \times \left( \frac{F \times z \times p}{R \times T} \right) \times \frac{\dot{V}}{I}.$$

Where $n_x$ is the measured mole fraction of the individual product measured by GC, $F$ the Faraday constant, $z$ the charge transfer number, $p$ the pressure, $R$ the ideal gas constant, $T$ the temperature, $\dot{V}$ the volume flow and $I$ the total current. The nitrogen flow rate is assumed to be constant and not to be affected by processes in the cell. The change of the nitrogen fraction of the product gas is used to calculate a

corrected volume flow to compensate for the reduction in volume flow due to $CO_2$ crossover. The partial current density of the product $x$ was calculated by multiplying the corresponding Faraday efficiency with the total current density

$$i_x = FE_x \times i.$$

## Neutron radiographic imaging

The neutron radiography measurements were performed at the NeXT instrument at the Institut Laue-Langevin[38]. The neutron beam has a white spectrum with wavelengths between approximately 1.5 Å and 20 Å. The through-plane direction of the assembled cell was oriented horizontally. To enhance the resolution in this horizontal direction, while maximizing flux, the beam aperture was set to a width of 5 mm and a height of 30 mm by a slit, yielding for the collimation ratio $L/D$ values of 2000 and 333 in the horizontal and vertical directions, respectively. The neutron imaging detector used in this experiment is an intensified neutron microscope from the ANTARES beamline of Forschungs-Neutronenquelle Heinz Maier-Leibnitz (FRM II)[39,40]. It uses a 5-μm-thick Gd$^{157}_2$O$_2$S scintillator that absorbs neutrons and emits scintillation light at green wavelengths. The microscope is built with two infinity corrected lenses facing each other, with a Zeiss 55 mm f/1.4 lens as the objective lens and a Nikon 70–200 mm f/2.8 zoom lens as the tube lens. A 1:1 relay is mounted between intensifier and camera. In all the experiments, the Nikon lens was set at $f = 200$ mm, yielding a magnification of 3.63. A Photonis 1-stage image intensifier (Cricket, 18 mm diameter) is used to amplify the scintillation light coming out of the light microscope and the 1:1 relay. The camera is a Hamamatsu Fusion BT sCMOS camera with 2304 × 2304 pixels, 6.5 μm in size. With the magnification of 3.63, the setup yields a maximum field of view of about 5.7*5.7 mm, with an effective pixel pitch of 1.79 μm and an estimated spatial resolution of about 6 μm. The exposure time of each neutron radiography is 10 s. For each neutron radiographic image gamma spot noise was removed[41]. Electrolysis cell images, open beam images (90 frames) and camera dark images (30 frames) were taken to calculate the normalized neutron radiographic images. The normalization of the neutron radiographic images was conducted as:

$$IMG_{norm} = \frac{IMG_{cell} - IMG_{dark}}{IMG_{open} - IMG_{dark}}.$$

Where $IMG_{norm}$ represents the normalized image; $IMG_{cell}$ represents the obtained radiography of the electrolysis cell; $IMG_{dark}$ represents the dark image of the detector; $IMG_{open}$ represents the open beam image taken without electrolysis cell. Each image presented in this work represents an average of the images taken in a 15 min current hold period (90 frames with 10 s exposure time), neglecting the first 100 s (10 frames) for cell equilibration and the period of 2 single condensate events observed at 100 mA cm$^{-2}$ and 300 mA cm$^{-2}$ (see Fig. 4).

To highlight the intensity changes at the different operating points, the images were referenced to an image of the zero current cell. This was done by dividing the averaged and normalized image of the particular current density ($IMG_i$) by the averaged and normalized image of the zero current cell ($IMG_0$):

$$IMG_i^{referenced} = \frac{IMG_i}{IMG_0}.$$

## Data availability

All data displayed in the figures of the manuscript and the supplementary information are provided in the Source Data file. The single averaged neutron radiographic images presented in Fig. 3 are provided

with this paper. The neutron radiography raw data of this study is available from the corresponding author upon request due to the large size. Source data are provided with this paper.

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

## Acknowledgements
This work was funded by the Vector Foundation ($CO_2$-to-X, S.V.) and the German Federal Ministry of Education and Research (NeutroSense, O5K19VFA, M.B. and 05K19WO2, M.S.). The authors want to thank the Institute Laue-Langevin (ILL) for hosting the neutron radiography experiment[42].

## Author contributions
S.V., M.B. and M.S. acquired project funding and supervised the project. J.D. planned the project with the help of S.K. and S.V. All electrochemical measurements were conducted by J.D.. M.S. and Y.H. performed the neutron radiography with the help of L.H. and A.T., who hosted the experiments at the NeXT instrument. L.B. helped with data analysis and image processing. J.D. mainly wrote the paper. Y.H. wrote most of the methods subsection "Neutron radiographic imaging". All authors discussed the results and commented on the manuscript.

## Funding

## Competing interests
The authors declare no competing interests.
