## [Peer Review File · Nature Communications]

Reviewers' comments:

Reviewer #1 (Remarks to the Author):

The Authors present an insight in a zero-gap CO₂ electrolyzer cell under operation, which is clearly a nice piece of work. The application of neutron imaging is well presented, but the conclusions drawn are not very new or convincing. The precipitate formation during operation was witnessed by several groups earlier, and the details of water management in the cell are not revealed here in details. Therefore, I feel that these achievements are more technical than novel, and hence I do not think that this paper belongs to Nature Communications. I believe it would be more suitable for more technical journals, such as ACS AMI, Applied Catalysis B, etc.

Some comments regarding the details of the manuscript:

Neutron imaging is a nice tool for visualizing the inner structure of the cell. Identifying precipitate, dry regions or bubbles is however very questionable, if not impossible. In Fig. S6 it is seen that K₂CO₃ is almost transparent for neutrons, and therefore its formation might result in the exact same image as a completely dry region. Assuming that the precipitate is hydrated is completely unfounded. If the water transport within the cell is to be described, other techniques should be applied to support these findings.

Regarding Fig. 1a, the followings are written: "As the cathode GDE is brighter in the land area of the cathode flow field even at 0 mA cm⁻², it can be assumed, that less precipitates are located in that area". Is there really any precipitate in the GDE at 0 current?

At higher current densities the membrane seems to be fairly dry. This would suggest a lower water concentration at the cathode surface, but we still see and increased HER at higher current densities. I find this a little contradictory.

"Going from 200 mA cm⁻² to 300 mA cm⁻² this trend for the cathode GDE is coming to halt and some regions of the GDE seem to get even slightly wetter again." – Why would the water content of the cathode GDE increase at higher current densities?

Instead of the Ti frit typically applied in zero gap cells, the authors used a carbon paper as anode support. Isn't this too hydrophobic to allow the proper wetting of the catalyst surface?

Reviewer #2 (Remarks to the Author):

The manuscript "High-resolution neutron imaging of carbonate precipitation and water transport in zero-gap CO₂ electrolysis" by J. Disch et al. is an interesting experimental attempt to visualize the transport of ions and molecules in a CO₂ electrolyzer by state-of-the-art neutron imaging, which is of utmost importance for the further optimization of such cells.

Thus, the topic and aims are relevant and of interest to the specialists working in CO₂-electrolysis as well as to the neutron imaging community. The research is well performed, and the research results justified (details follow later). Thus I have no problem in recommending publication of this nice piece of work. However, the appetite comes with eating. Roughly speaking, the authors discuss one "beamtime experiment", which should be elaborated and be put in perspective of the science and technology needed in CO₂ electrolysis, in particular the importance of mass transport. Adding a discussion around this may make this paper a cornerstone in CO₂ electrolysis. Concretely:

- A figure of the CO₂ electrolyzer showing the electrodes and membrane geometry and what the neutrons probe, and highlighting the processes taking place. For understanding it is helpful to pinpoint, where water and CO₂ is consumed, where a pH gradient builds up, where the gases evolve, etc., and how potential precipitations may perturb the overall process. I have the impression that the authors involved have some background in water electrolysis, where similar processes such as water management are relevant. However, CO₂ electrolysis is more complex simply because CO₂ is involved (see, e.g., Ref. 37; and Nesbitt et al., ACS Catal. 10, 14093–14106 (2020); Lu et al.,

J. Am. Chem. Soc. 142, 15438–15444 (2020); Borgschulte et al., Front. Energy Res. 9:784082 (2022)). Figure 1 gives some of the aforementioned reactions, but it may be split in two figures one only showing the sketch (and critical phenomena). Also, to generalize the findings. The authors correctly mention that the findings may not be applicable to other electrolyzer setups (line 77 ff). Here, discussion/explanations are most important (e.g., what are the differences).

- The attribution of the neutron contrast to K(H)CO₃ is the most important finding, but also the most critical one. It is based on a kind of of handwaving arguments, but this is for sure not a chemical identification. Spatially resolved spectroscopy methods such as Raman spectroscopy (Lu et al., J. Am. Chem. Soc. 142, 15438–15444 (2020)) could provide evidence for this argumentation. I recommend to apply this (I do not know whether possible in this short time).

Response to referees

Dear Sir or Madam,

We thank the editor for considering our manuscript for review and the referees for their comments. The questions and comments from the referees lead to fruitful discussions and helped to improve the manuscript. Below, we address each comment separately and highlight the changes we made to the manuscript.

Reviewer #1

Comment 1:

The Authors present an insight in a zero-gap CO₂ electrolyzer cell under operation, which is clearly a nice piece of work. The application of neutron imaging is well presented, but the conclusions drawn are not very new or convincing. The precipitate formation during operation was witnessed by several groups earlier, and the details of water management in the cell are not revealed here in details. Therefore, I feel that these achievements are more technical than novel, and hence I do not think that this paper belongs to Nature Communications. I believe it would be more suitable for more technical journals, such as ACS AMI, Applied Catalysis B, etc.

Response:

We thank the reviewer for the profound feedback. We tried to significantly work on our manuscript, including new ex-situ characterization results which hopefully back up our findings in a convincing way (see below).

Comment 2:

Neutron imaging is a nice tool for visualizing the inner structure of the cell. Identifying precipitate, dry regions or bubbles is however very questionable, if not impossible.

Response:

The reviewer is right, that neutron imaging is a complex method, and any results need to be interpreted with care. However, it was repeatedly proven over more than a decade now that neutron imaging is especially suitable for the in-situ investigation of water distribution and bubble formation in electrochemical devices, what has been proven in a multitude of publications¹⁻⁷. The identification of precipitates by neutron imaging is less explored and therefore discussed in detail in the comments following below.

Comment 3:

In Fig. S6 it is seen that K₂CO₃ is almost transparent for neutrons, and therefore its formation might result in the exact same image as a completely dry region.

Response:

Yes, this is right. This is why we did not claim that K_2CO_3 is imaged directly in the neutron radiography. In fact, we stated that K_2CO_3 can only be imaged indirectly by imaging the retained water (original manuscript line 172ff) and specifically added the mentioned Fig. S6 to illustrate that. However, Fig. S6 also shows that $KHCO_3$ can significantly contribute to the neutron attenuation, which makes imaging of $KHCO_3$ possible (see comment 7 and comment 4 of Reviewer #2).

Considering the reviewer's comment, we concluded that the wording in the manuscript and its title could be more precise to not be misunderstood. We therefore changed the title and some expressions in the manuscript.

Changes to the manuscript:

Title, line 1f: **High-resolution neutron imaging of salt precipitation and water transport in zero-gap CO_2 electrolysis**

Line 22f: Neutron imaging further shows higher **salt** accumulation under the cathode channel of the flow field compared to the land.

Line 31f: **Salt precipitation and associated electrode flooding** are observed in neutron images visualizing the cause of commonly observed decreasing Faraday efficiency.

Line 333ff: Although precipitates **and associated electrode flooding** are visible in the neutron images over the entire current range (precipitates potentially already formed during pretests), the cell is demonstrating good cell performance with FE_{CO} above 90 % for current densities below 300 mA cm^{-2} .

Comment 4:

Assuming that the precipitate is hydrated is completely unfounded. If the water transport within the cell is to be described, other techniques should be applied to support these findings.

Response:

Carbonate precipitates are normally porous and hydrophilic and hence increase the capillary pressure⁸. Considering the deliquescent properties of carbonates and the moist reaction environment (humidified cathode gas stream, aqueous anolyte, a membrane with a high water uptake and ambient temperature) carbonates are very likely hydrated. This should be the case, especially in the zero current state, where no water is consumed by the electrode reactions. Fig. S6 also shows that already small amounts of water have a significant effect on the neutron transmission. However, the cathode GDE land region stays significantly brighter than the channel region at 0 mA/cm^2 . It has been shown in literature, that the relative humidity is quite high in the cathode compartment even at higher current densities. Particularly the publication of Wheeler et al. is to mention in this context,

in which the humidity in the cathode flow field is measured at 4 different positions, with and without humidification of the gas feed⁹. They show that without prior gas humidification the gas stream is humidified inside the cell, even at 200 mA / cm² (RH of 79% at the outlet, also using similar Sustainion membranes).

We now address the humid environment in the manuscript.

Changes to the manuscript:

Line 195ff: K₂CO₃ shows significantly lower neutron absorption and scattering rates than water (see SI Figure S6 for neutron transmission rates). **Due to its low neutron attenuation, K₂CO₃ can only be observed indirectly, by imaging the retained water. Considering the deliquescent properties of K₂CO₃ and the moist reaction environment (humidified cathode gas stream, aqueous anolyte, a membrane with a high water uptake¹⁰ and ambient temperature) carbonates inside the cell are very likely hydrated⁸. This should be the case, especially in the zero current state, when no water is consumed by the cathode reactions and condensing water in the cathode flow field is observed.** KHCO₃ instead contains a hydrogen atom in its structure, yielding higher interactions with neutrons than K₂CO₃.

Comment 5:

Regarding Fig. 1a, the followings are written: “As the cathode GDE is brighter in the land area of the cathode flow field even at 0 mA cm⁻², it can be assumed, that less precipitates are located in that area”. Is there really any precipitate in the GDE at 0 current?

Response:

As described in detail in the SI (line 69ff) and mentioned in the manuscript (line 265f), there have been preceding electrochemical measurements to the experiments reported in the manuscript. Therefore, we see no contradiction with precipitates or highly saturated solution in the cathode GDE at zero current. We now do also mention it in that section.

Changes to the manuscript:

Line 214ff: **For example**, as the cathode GDE is brighter in the land area of the cathode flow field even at 0 mA cm⁻², it can be assumed, that less precipitates are located in that area **(note that KHCO₃ and K₂CO₃ might have accumulated in the cathode GDE already in the preceding electrochemical measurements described in the SI).**

Comment 6:

At higher current densities the membrane seems to be fairly dry. This would suggest a lower water concentration at the cathode surface, but we still see and increased HER at higher current densities. I find this a little contradictory.

Response:

The reviewer is right, that this might at first seem counterintuitive. Nonetheless, several points have to be considered:

- Figure 5 shows that the membrane is getting slightly dryer with rising current density compared to the 0 current image. The change in water content, however, is not as drastic as in other regions in the cell (see e.g. cathode GDE, condensate in cathode flow field channels). This is a reasonable result, as the utilized Sustainion membrane was reported to have a very high water uptake (~80 % reported by Lindquist et al.)^{10,11}. Thereby it provides liquid water to the cathode catalyst layer as it is in direct contact with it. Moreover, in dry cathode AEM water electrolysis, significantly higher HER current densities are achieved, only providing water from the anode side through the membrane^{12,13}.
- Further, as Reviewer #2 mentioned, mass transport plays an important role in CO₂ electrolysis. Most likely, more limiting than water availability is the availability of CO₂ at the cathode. CO₂ has to reach the active sites of the cathode catalyst layer by diffusing from the cathode flow field channel through the gas diffusion layer, the micro porous layer and finally the pores of the catalyst layer. With rising current density more CO₂ is consumed by the CO₂ reduction reaction and the carbonation reaction, lowering the CO₂ concentration at the electrode surface, while there is still enough water to maintain the HER (concentration of CO₂ in a humidified gas stream C_{CO2} = 0.041 M compared to concentration of pure liquid water C_{H2O} = 55.5 M¹⁴). Precipitates in the electrode add an additional diffusion barrier for CO₂.

Changes to the manuscript:

Line 289ff: While this might be surprising, one has to consider that a region depleted of CO₂ might still have a sufficiently high water content for the hydrogen evolution reaction. **The utilized Sustainion® membranes were reported to have a very high water uptake of ~80 %^{10,11} and are only getting slightly drier during cell operation (see Figure 5), directly providing water to cathode catalyst layer. CO₂, however, has to reach the active sites of the cathode catalyst layer by diffusing from the cathode flow field channel through the gas diffusion layer, the micro porous layer and finally the pores of the catalyst layer. With rising current density more CO₂ is consumed by the CO₂ reduction reaction and the carbonation reaction, lowering the CO₂ concentration at the electrode surface. Precipitates in the electrode add an additional diffusion barrier for CO₂ (see Figure 1d). Thus the reaction shifts to the HER resulting in lower faradaic efficiency of the CO₂ reduction at high current densities (Figure 2c) instead of increasing the overpotential.**

Comment 7:

“Going from 200 mA cm⁻² to 300 mA cm⁻² this trend for the cathode GDE is coming to halt and some regions of the GDE seem to get even slightly wetter again.” – Why would the water content of the cathode GDE increase at higher current densities?

Response:

We agree with the reviewer that this observation is not very intuitive, as the water consumption on the cathode side rises with increasing current density. Nonetheless, we observed this behavior (also in other measurements). In our manuscript we explain

possible reasons for this observation (line 223ff). The intensity decrease is mainly observed in the cathode gas diffusion layer and more pronounced in the channel region of the cathode flow field (see Figure 4 and 5). As this is a continuous process (see Figure 5) we conclude that this intensity decrease is most probably caused by KHCO_3 precipitates (and water retained by the precipitates) accumulating in that area.

To additionally support this statement, we conducted SEM/EDX measurements of a cross-section of the cathode GDE and Raman measurements of the silver catalyst layer after cell operation and added them to the Supplementary Information (see below).

The SEM/EDX measurements clearly show potassium containing precipitates in the silver catalyst layer, the micro porous layer and in the gas diffusion layer of the cathode.

The Raman measurements further confirm the presence of KHCO_3 in the silver catalyst layer. The spectrum does not show the typical bands for K_2CO_3 .

Different publications concluded from post-experiment analysis that the precipitates consist of a mixture of different carbonic salts:

E.g. Endrődi et al. conducted XRD measurements and concluded that the precipitated salts consist of a mixture of KHCO_3 and $\text{K}_4\text{H}_2(\text{CO}_3)_3 \cdot 1.5\text{H}_2\text{O}$, with an approximate 1:3 weight ratio¹⁵. Xu et al. showed with Raman measurements that in their case, the precipitates consist of a mixture of K_2CO_3 and KHCO_3 with more K_2CO_3 than KHCO_3 ¹⁶. All studies report species containing hydrogen atoms in the precipitates (thus, species yielding significant neutron interaction). The ratio of the different species might strongly differ in different cells, depending on material properties and operating conditions.

Changes to the manuscript:

Line 205ff: Nonetheless, KHCO_3 is also strongly hygroscopic and retains water, which might more drastically affect its neutron opacity in a humid environment. **Scanning electron microscopy and energy dispersive X-ray spectroscopy (SEM/EDX) measurements of a cathode cross section further show precipitates containing potassium in the silver catalyst layer, the micro porous layer and in the gas diffusion layer (see SI Figure S7). The presence of KHCO_3 in the catalyst layer was additionally verified by post-operation Raman measurements (see SI Figure S8). Considering those results and the mentioned observations made by other groups^{15,17}, it can be assumed that the dark dents originate from precipitated KHCO_3 and water retained by the precipitates in that area.**

By that, the intensity differences alongside the cathode GDE allow to deduce where precipitates are mainly located in the electrode. For example, as the cathode GDE is brighter in the land area of the cathode flow field even at 0 mA cm^{-2} , it can be assumed, that less precipitates are located in that area.

Line 237ff: Going from 200 mA cm^{-2} to 300 mA cm^{-2} this trend for the cathode GDE is coming to halt and **the intensity in some regions of the cathode gas diffusion layer even decreases.**

Line 252ff: However, considering the afore-mentioned points and the post-operation Raman and SEM/EDX measurements (see SI Figure S7 and S8), it is most probable that the accumulation of KHCO_3 causes the observed drop in intensity.

Line 272ff: Again, if the change in intensity would only account for water content, this observation would be counterintuitive, as the increased reaction rate leads to higher water consumption and electro-osmotic drag from cathode to anode (see Figure 1a). This observation in the cathode GDL suggest that KHCO_3 is continuously accumulating during cell operation. The different slopes of the cathode GDL intensity curve translate to KHCO_3 accumulation already starting at 100 mA cm^{-2} and significantly increasing at 300 mA cm^{-2} .

Changes to the Supplementary Information:

S 1 Scanning electron micrographs and energy dispersive X-ray spectroscopies of the cathode electrode cross section after cell operation. a) electron micrograph and elemental maps of carbon, fluorine, oxygen, silver, potassium and iridium, b) a layered image of the electron micrograph and the elemental map of potassium, c) an electron micrograph layered with the elemental maps of iridium, silver and fluorine, identifying the catalyst layers and the hydrophobic microporous layer, and d) spectrum obtained from the energy dispersive X-ray spectroscopy. The cross section was prepared by cryo-cutting.

S 2 Raman spectra of pure potassium bicarbonate and of the cathode catalyst layer after cell operation. The spectra were recorded at different positions of the cathode GDE catalyst layer surface after carefully removing the membrane. The phenyl group signal results from the Sustinion cathode catalyst binder ^{18,19}. Raman spectra were obtained using a WITec alpha 300 confocal Raman microscope with a 532 nm laser operated at 10 ± 1 mW as the excitation source. Average spectra of the samples were produced by averaging five single spectra from each respective sample. All single spectra were integrated for 0.5 s and accumulated ten times. Background subtraction and fitting was done using WITec project.

Comment 8:

Instead of the Ti frit typically applied in zero gap cells, the authors used a carbon paper as anode support. Isn't this too hydrophobic to allow the proper wetting of the catalyst surface?

Response:

We decided to use this anode as it is a commercially available state-of-the-art electrode and has been proven to be suitable for this application in several studies ^{19,20, 21}. By choosing commercially available materials, we try to make our results reproducible for other groups. As the neutron radiographic images show electrolyte fully enters the anode GDE. Thus, the carbon paper works well as a substrate despite its wettability.

Reviewer #2:

Comment 1:

The manuscript "High-resolution neutron imaging of carbonate precipitation and water transport in zero-gap CO₂ electrolysis" by J. Disch et al. is an interesting experimental attempt to visualize the transport of ions and molecules in a CO₂ electrolyzer by state-of-the-art neutron imaging, which is of utmost importance for the further optimization of such cells.

Thus, the topic and aims are relevant and of interest to the specialists working in CO₂-electrolysis as well as to the neutron imaging community. The research is well performed, and the research results justified (details follow later). Thus I have no problem in recommending publication of this nice piece of work. However, the appetite comes with eating. Roughly speaking, the authors discuss one "beamtime experiment", which should be elaborated and be put in perspective of the science and technology needed in CO₂ electrolysis, in particular the importance of mass transport. Adding a discussion around this may make this paper a cornerstone in CO₂ electrolysis.

Response:

We thank the reviewer for the very positive evaluation of our manuscript. Within the restricted time budget of a revision process, we tried our best to further back up our statements following the reviewer's recommendations. Therefore, we included now further post-mortem microanalysis of the MEAs by SEM/EDX and Raman. The measurements clearly confirm the presence of potassium accumulations in the cathode.

Comment 2:

Concretely:

- A figure of the CO₂ electrolyzer showing the electrodes and membrane geometry and what the neutrons probe, and highlighting the processes taking place. For understanding it is helpful to pinpoint, where water and CO₂ is consumed, where a pH gradient builds up, where the gases evolve, etc., and how potential precipitations may perturb the overall process. I have the impression that the authors involved have some background in water electrolysis, where similar processes such as water management are relevant. However, CO₂ electrolysis is more complex simply because CO₂ is involved (see, e.g., Ref. 37; and Nesbitt et al., ACS Catal. 10, 14093–14106 (2020); Lu et al., J. Am. Chem. Soc. 142, 15438–15444 (2020); Borgschulte et al., Front. Energy Res. 9:784082 (2022)).

Figure 1 gives some of the aforementioned reactions, but it may be split in two figures one only showing the sketch (and critical phenomena). Also, to generalize the findings.

Response:

We thank the reviewer for the feedback and the proposed amendments. Considering the propositions we revised Figure 1. We added a Figure 1c and 1d, illustrating the idealized processes at the cathode GDE and the influence of carbonates on the CO₂ mass transport. Additionally, we added Figure S1b to the supplementary information, showing a rendering of the neutron detector and a cross section of the cell fixture, to provide a better understanding what the neutrons probe.

Changes to the manuscript:

Line 63ff: Although the AEM hinders potassium cations from crossing the membrane in bigger quantities by Donnan exclusion, a small amount of cations is still migrating through the membrane. As cations move from the anode to the cathode side, they drag water in their hydration shell with them. **Over time potassium, carbonate and bicarbonate ions can accumulate at the cathode. When exceeding the solubility limit, by increasing ion concentration or by consumption of water, salt crystallites can precipitate on the cathode**

side. Figures 1c and 1d illustrate how precipitates supposedly influence the processes in the cathode gas diffusion electrode (GDE). Carbonate precipitates are normally porous and hydrophilic, increasing the capillary pressure inside the cathode GDE⁸. Hence, the precipitates make the GDE prone to electrode flooding. Liquids and precipitates can clog pores in the catalyst layer, micro porous layer and the gas diffusion layer of the cathode GDE, and thereby impede the gas transport²². With an increasing amount of precipitates, the mass transport resistance for the CO₂ to reach the active sites of the catalyst layer increases and the CO₂ concentration at the catalyst surface will eventually decrease. The lower CO₂ concentration may cause a selectivity shift towards the HER, as water can still be sufficiently supplied from the anode side through the membrane.

Figure 1 Transport processes and cell components. a) water management highlighted in schematic of reactions and transport processes during operation, b) enlarged section of a normalized neutron radiographic image of the CO₂ electrolysis cell, c) idealized cathode processes, and d) influence of precipitates on the cathode processes.

Changes to the Supplementary Information:

S 3 Cell fixture and neutron detector/cell alignment. a) an explosion view of the custom-built electrolysis cell (End plates: Stainless steel, flow fields: Au coated Ti grade 2, gaskets: PTFE), and b) a rendering of the neutron detector and of the cell fixture cross section in alignment to the incident neutron beam. The scintillator sits in the neutron detector head close to the cell fixture. The emitted scintillation light is reflected to the detector, which is orientated perpendicular to the incident neutron beam.

Comment 3:

The authors correctly mention that the findings may not be applicable to other electrolyzer setups (line 77 ff). Here, discussion/explanations are most important (e.g., what are the differences).

Response:

The reviewer raised a fair point and we therefore added that discussion to the manuscript.

Changes to the manuscript:

Line 83ff: The technological advances of neutron imaging detectors significantly improved the resolution of neutron radiography in recent years. Nonetheless, only two neutron-imaging studies investigating CO₂ electrolyzers have been published up to now. The first study investigated bubble formation in a liquid buffer layer on the cathode side³. Besides high cell potentials, cells with a liquid catholyte between the membrane and cathode GDE

often exhibit fast electrode flooding, as the cathode is in direct contact with the liquid electrolyte. The circulating electrolyte, however, can wash away precipitates and depending on its chemical composition change the ion concentration at the cathode side compared to a zero-gap cell. The second study conducted neutron imaging of a zero-gap cell with a cation exchange membrane²³. Zero-gap cells using cation exchange membranes show significantly lower CO selectivity compared to cells using AEMs, as they provide an acidic reaction environments to the cathode, facilitating the competing HER²⁴. Furthermore, protons moving from the anode to the cathode are responsible for the ionic charge transfer between the anode and cathode. The ionic transport and the electroosmotic drag are directed in the opposite direction compared to cells using AEMs, drastically changing the water balance. As the water balance and flooding behavior in the above-mentioned studies strongly differ, their results cannot be applied to AEM-based zero-gap CO₂ electrolyzers. Etzold et al. have recently highlighted this lack of neutron studies for zero-gap CO₂ electrolysis in a review²⁵.

Comment 4:

The attribution of the neutron contrast to K(H)CO₃ is the most important finding, but also the most critical one. It is based on a kind of handwaving arguments, but this is for sure not a chemical identification. Spatially resolved spectroscopy methods such as Raman spectroscopy (Lu et al., J. Am. Chem. Soc. 142, 15438–15444 (2020)) could provide evidence for this argumentation. I recommend to apply this (I do not know whether possible in this short time).

Response:

The reviewer is right, that it is not possible to quantitatively deconvolute the contribution of KHCO₃ and water to the neutron attenuation with our method. Unfortunately, the proposed in situ Raman spectroscopy is not only impossible in this short time, but is not possible at all, as we investigate a zero-gap cell architecture (membranes and electrodes are not transparent for the Raman laser). Lu et al. only investigate the concentration gradients inside the liquid catholyte and cannot analyze the gradients inside the cathode GDE. The liquid catholyte renders the results not applicable to a zero-gap cell (also see comment 3, different cell architectures).

However, we followed the reviewer's recommendation to employ Raman spectroscopy as method of choice, and backed the findings up with additional SEM/EDX measurements of the cathode electrode to further investigate the distribution and the composition of the precipitates (see comment 7 of Reviewer #1).

The SEM/EDX measurements clearly show potassium containing precipitates in the silver catalyst layer, the micro porous layer and in the gas diffusion layer. The Raman measurements further prove the presence of KHCO₃ in the cathode.

Considering the above-mentioned, a closer look at Figure S6 can help to further clarify the interpretation of the neutron radiography data. As Reviewer #1 correctly stated, the contribution of K₂CO₃ to the neutron attenuation can be neglected, whereas KHCO₃ shows significant neutron attenuation:

S 4 Calculated neutron transmission for different compounds and sample thicknesses calculated with the online calculator from “NIST Center for Neutron Research” [<https://www.ncnr.nist.gov/resources/activation/>, Date: 02.2022]. The wavelength was set to 3 Å.

The neutron attenuation of water is still significantly higher than that of KHCO₃. In the zero current state and at low current densities (low water consumption) the neutron attenuation might be dominated by water. At higher current densities (increased water consumption and electroosmotic drag) the contribution of KHCO₃ to the neutron attenuation in the cathode GDE gets more significant. At 200 mA cm⁻² the liquid water in the cathode flow field has already fully dried off, however, the precipitates are clearly visible.

All information we gained from the Raman measurements, SEM/EDX measurements and literature (see comment 7 of Reviewer #1 and line 191ff in the manuscript) supports the claim that KHCO₃ contributes to the neutron attenuation in the presented radiographic images.

Changes to the manuscript:

Line 188ff: Nevertheless, at 300 mA cm⁻² these bulges are still clearly visible, **whereas the liquid condensate in the channel has fully dried off**. Even though the images are averaged over several minutes, the bulges show sharp contours, indicating an immobile solid phase.

All other related changes are listed in the response to comment 7 of Reviewer #1.

Concluding remarks

Further minor changes to the manuscript that are not mentioned in this letter are highlighted in the newly submitted manuscript file. We appreciate the detailed revision of our manuscript. We hope that we could answer all questions to the satisfaction of the referees. Furthermore, we are open to discuss any further questions.

Sincerely yours,

Severin Vierrath

References

1. Kardjilov, N., Manke, I., Woracek, R., Hilger, A. & Banhart, J. Advances in neutron imaging. *Materials Today* **21**, 652–672; 10.1016/j.mattod.2018.03.001 (2018).
2. Boillat, P., Frei, G., Lehmann, E. H., Scherer, G. G. & Wokaun, A. Neutron Imaging Resolution Improvements Optimized for Fuel Cell Applications. *Electrochem. Solid-State Lett.* **13**, B25; 10.1149/1.3279636 (2010).
3. Krause, K. *et al.* Electrolyte layer gas triggers cathode potential instability in CO₂ electrolyzers. *Journal of Power Sources* **520**, 230879; 10.1016/j.jpowsour.2021.230879 (2022).
4. Lehmann, E. H. & Wagner, W. Neutron imaging at PSI. A promising tool in materials science and technology. *Appl. Phys. A* **99**, 627–634; 10.1007/s00339-010-5606-3 (2010).
5. Hoeh, M. A. *et al.* In-Operando Neutron Radiography Studies of Polymer Electrolyte Membrane Water Electrolyzers. *ECS Trans.* **69**, 1135–1140; 10.1149/06917.1135ecst (2015).
6. Omasta, T. J. *et al.* Beyond catalysis and membranes. Visualizing and solving the challenge of electrode water accumulation and flooding in AEMFCs. *Energy Environ. Sci.* **11**, 551–558; 10.1039/C8EE00122G (2018).
7. Zlobinski, M., Schuler, T., Büchi, F. N., Schmidt, T. J. & Boillat, P. Transient and Steady State Two-Phase Flow in Anodic Porous Transport Layer of Proton Exchange Membrane Water Electrolyzer. *J. Electrochem. Soc.* **167**, 84509; 10.1149/1945-7111/ab8c89 (2020).
8. Li, M. *et al.* The role of electrode wettability in electrochemical reduction of carbon dioxide. *J. Mater. Chem. A* **9**, 19369–19409; 10.1039/d1ta03636j (2021).
9. Wheeler, D. G. *et al.* Quantification of water transport in a CO₂ electrolyzer. *Energy Environ. Sci.* **13**, 5126–5134; 10.1039/d0ee02219e (2020).
10. Kaczur, J. J., Yang, H., Liu, Z., Sajjad, S. D. & Masel, R. I. Carbon Dioxide and Water Electrolysis Using New Alkaline Stable Anion Membranes. *Frontiers in chemistry* **6**, 263; 10.3389/fchem.2018.00263 (2018).
11. Lindquist, G. A. *et al.* Performance and Durability of Pure-Water-Fed Anion Exchange Membrane Electrolyzers Using Baseline Materials and Operation. *ACS applied materials & interfaces*; 10.1021/acsami.1c06053 (2021).
12. Cho, M. K. *et al.* Alkaline anion exchange membrane water electrolysis. Effects of electrolyte feed method and electrode binder content. *Journal of Power Sources* **382**, 22–29; 10.1016/j.jpowsour.2018.02.025 (2018).
13. Parrondo, J., George, M., Capuano, C., Ayers, K. E. & Ramani, V. Pyrochlore electrocatalysts for efficient alkaline water electrolysis. *J. Mater. Chem. A* **3**, 10819–10828; 10.1039/C5TA01771H (2015).

14. Weekes, D. M., Salvatore, D. A., Reyes, A., Huang, A. & Berlinguette, C. P. Electrolytic CO₂ Reduction in a Flow Cell. *Accounts of chemical research* **51**, 910–918; 10.1021/acs.accounts.8b00010 (2018).
15. Endródi, B. *et al.* Operando cathode activation with alkali metal cations for high current density operation of water-fed zero-gap carbon dioxide electrolyzers. *Nature energy* **6**, 439–448; 10.1038/s41560-021-00813-w (2021).
16. Xu, Y. *et al.* Self-Cleaning CO₂ Reduction Systems. Unsteady Electrochemical Forcing Enables Stability. *ACS Energy Lett.*, 809–815; 10.1021/acsenergylett.0c02401 (2021).
17. Kong, Y. *et al.* Visualisation and quantification of flooding phenomena in gas diffusion electrodes (GDEs) used for electrochemical CO₂ reduction. A combined EDX/ICP–MS approach (2022).
18. Nwabara, U. O. *et al.* Binder-Focused Approaches to Improve the Stability of Cathodes for CO₂ Electroreduction. *ACS Appl. Energy Mater.* **4**, 5175–5186; 10.1021/acsaem.1c00715 (2021).
19. Kutz, R. B. *et al.* Sustainion Imidazolium-Functionalized Polymers for Carbon Dioxide Electrolysis. *Energy Technol.* **5**, 929–936; 10.1002/ente.201600636 (2017).
20. Larrazábal, G. O. *et al.* Analysis of Mass Flows and Membrane Cross-over in CO₂ Reduction at High Current Densities in an MEA-Type Electrolyzer. *ACS applied materials & interfaces* **11**, 41281–41288; 10.1021/acsami.9b13081 (2019).
21. Liu, Z., Yang, H., Kutz, R. & Masel, R. I. CO₂ Electrolysis to CO and O₂ at High Selectivity, Stability and Efficiency Using Sustainion Membranes. *J. Electrochem. Soc.* **165**, J3371–J3377; 10.1149/2.0501815jes (2018).
22. Leonard, M. E., Clarke, L. E., Forner-Cuenca, A., Brown, S. M. & Brushett, F. R. Investigating Electrode Flooding in a Flowing Electrolyte, Gas-Fed Carbon Dioxide Electrolyzer. *ChemSusChem*; 10.1002/cssc.201902547 (2019).
23. Shafaque, H. W. *et al.* Boosting Membrane Hydration for High Current Densities in Membrane Electrode Assembly CO₂ Electrolysis. *ACS applied materials & interfaces*; 10.1021/acsami.0c14832 (2020).
24. Delacourt, C., Ridgway, P. L., Kerr, J. B. & Newman, J. Design of an Electrochemical Cell Making Syngas (CO+H₂) from CO₂ and H₂O Reduction at Room Temperature. *J. Electrochem. Soc.* **155**, B42; 10.1149/1.2801871 (2008).
25. Etzold, B. J.M. *et al.* Understanding the activity transport nexus in water and CO₂ electrolysis. State of the art, challenges and perspectives. *Chemical Engineering Journal* **424**, 130501; 10.1016/j.cej.2021.130501 (2021).

REVIEWERS' COMMENTS

Reviewer #1 (Remarks to the Author):

It is clear, that the Authors worked a lot on the manuscript, that is now much more clear. Reading it again I accept the novelty in the neutron imaging and in the spatial resolution of precipitate formation and water transport. There are still some open questions, that will hopefully be addressed in the follow-up studies from the authors.

All things considered, I suggest the publication of the manuscript in its current form. As my questions were fully addressed, I do not have any questions/comments.

Reviewer #2 (Remarks to the Author):

My first review was already quite positive, with the only major concern about the chemical identification by neutrons. This is now supported by Raman spectroscopy, unfortunately not operando, but I think the results are sound nevertheless. I am happy! If I my help countering the first reviewer: he is of course right that the precipitation was already observed before. You might stress the point that in-situ imaging is very good tool in optimizing cell geometry and operation conditions ("an image is worth 1000 words, and a movie..."), less so in finding a new effect. This was also my intention of the suggestion to add a scheme of the processes involved. On the images one gets the feeling what to improve (e.g., the land/free space ratio).